# Trust and experiences of National Health Service healthcare do not fully explain demographic disparities in coronavirus vaccination uptake in the UK: a cross-sectional study

Daniel Allington ![ORCID] ,[1] Siobhan McAndrew,[2] Bobby Duffy,[3] Vivienne Moxham-Hall[3]

[1]Department of Digital Humanities, King's College London, London, UK
[2]Sheffield Methods Institute, University of Sheffield, Sheffield, UK
[3]Policy Institute, King's College London, London, UK

**Correspondence to**
Dr Daniel Allington;
daniel.allington@kcl.ac.uk

## ABSTRACT

**Objective** To test whether demographic variation in vaccine hesitancy can be explained by trust and healthcare experiences.

**Design** Cross-sectional study.

**Setting** Data collected online in April 2021.

**Participants** Data were collected from 4885 UK resident adults, of whom 3223 had received the invitation to be vaccinated against the novel coronavirus and could therefore be included in the study. 1629 included participants identified as female and 1594 as male. 234 identified as belonging to other than white ethnic groups, while 2967 identified as belonging to white ethnic groups.

**Primary and secondary outcome measures** Uptake of coronavirus vaccination.

**Results** Membership of an other than white ethnic group (adjusted OR (AOR)=0.53, 95% CI 0.35 to 0.84, p=0.005) and age (AOR=1.61, 95% CI 1.39 to 1.87, p<0.001 for a 1 SD change from the mean) were the only statistically significant demographic predictors of vaccine uptake. After controls for National Health Service (NHS) healthcare experiences and trust in government, scientists and medical professionals, the effect associated with membership of an other than white ethnic group appears more marginal (AOR=0.61, 95% CI 0.38 to 1.01, p=0.046), while the effect associated with age remains virtually unchanged. Exploratory analysis suggests that NHS healthcare experiences mediate 24% (95% CI 8% to 100%, p=0.024) of the association between ethnicity and uptake, while trust mediates 94% (95% CI 56% to 100%, p=0.001) of the association between NHS healthcare experiences and uptake.

**Conclusions** Members of other than white ethnic groups report inferior NHS healthcare experiences, potentially explaining their lower reported trust in government, scientists and medical professionals. However, this does not fully explain the ethnic gap in coronavirus vaccination uptake.

## INTRODUCTION

The problem of vaccine hesitancy, defined as 'delay in acceptance or refusal of vaccination despite availability of vaccination services',[1] has been recognised as a major global health

### Strengths and limitations of this study

► The sample was large, providing high statistical power, and was designed for representativeness of the UK adult population.

► Standard measures of trust and healthcare experiences were used, providing confidence that these variables have been measured robustly.

► It was not possible to obtain a probability sample and there was limited representation within the sample both of people who did not take up the invitation to be vaccinated and of members of other than white ethnic groups (given that both of these are relatively small minorities in the overall UK population).

► Moreover, given the younger age profile of other than white ethnic groups in the UK, it is difficult to disentangle the effects associated with age from the effects associated with ethnicity.

problem for several years.[2 3] However, it gained particular importance during the COVID-19 public health emergency: early in the pandemic, it was estimated that as much as 82% of a population might need vaccination or natural immunity against the disease before herd immunity could be reached,[4] but one survey indicated that less than 72% of the British population might be willing to accept vaccination against SARS-CoV-2, the novel coronavirus which causes COVID-19.[5] Multiple surveys have found demographic disparities in British respondents' expressed likelihood of vaccine uptake, with younger, less educated respondents, women, members of lower-income households and members of other than white ethnic groups all tending to express a lower probability of accepting the offer of vaccination.[6–8] The finding with regard to ethnicity has been of particular concern, given higher fatality rates for COVID-19 among members of minority

ethnic groups.[9] Actual uptake has been far higher than these surveys would have predicted, with one longitudinal study finding that 99% of those who had stated in December 2020 that they would accept vaccination and 86% of those who had stated at the same point that they would not or were unsure stated in February 2021 either that they would accept vaccination or that they had already done so.[10] However, despite this encouraging trend, demographic disparities appeared to persist,[11] although at a reduced level, with youth and membership of other than white ethnic groups now being the strongest predictors of coronavirus vaccine hesitancy uptake in the UK,[10] such that by 13 January 2021, 42.5% of white people aged over 80 but not residents of care homes, but only 20.5% of black people in the same category, had been vaccinated.[12] The potential health consequences of incomplete vaccination coverage were highlighted by the official claim that the majority of people hospitalised as a result of a COVID-19 outbreak in the English town of Bolton were unvaccinated yet eligible to be vaccinated.[13]

The possibility that lower trust in government, scientists and the medical system might lead to reduced coronavirus vaccine uptake among members of minority ethnic communities had already been noted before any survey data had been collected that might serve to test such a hypothesis.[14] Indeed, many studies have found a relationship between vaccine hesitancy and mistrust in medicine and other relevant institutions, both at the individual level[15–18] and at the national level,[19] with regard to vaccination against other diseases. Evidence now exists to suggest that coronavirus vaccine hesitancy among African Americans may be driven by mistrust in the medical establishment (itself resulting from awareness of past mistreatment and unethical practices on the part of medical professionals and researchers), as well as by perceptions of racism in the political system.[20] A systematic review of factors potentially explaining coronavirus vaccination uptake within minority ethnic communities in the UK found one study which identified '[l]ogistical and practical barriers, such as the location of vaccine centres and having to use public transport' as a potential driver of lower vaccine uptake rates, six which attributed reduced uptake to 'mistrust including pre-existing lower scientific or medical trust, conspiracy suspicions and attitudes', and four which found that 'people from minority ethnic backgrounds were more likely than White British groups to have received misinformation encouraging them not to have the vaccine'.[21]

Some experts have attributed heightened levels both of coronavirus misinformation and of coronavirus vaccine hesitancy among members of some minority ethnic communities in the UK to mistrust, which they argue is in turn driven by experiences of racism, discrimination and exclusion.[22] As members of minority ethnic groups report lower patient satisfaction and worse healthcare experiences in the UK,[23] and at an international level appear to suffer from 'higher incidences of healthcare associated infections, dosing errors, [adverse drug events], and

complications resulting from their care',[24] it would seem plausible that systematically poorer healthcare experiences may act as a driver of ethnic disparities in coronavirus vaccine uptake, alongside strategic and situational trust. Moreover, it would not appear outside the realms of possibility that some factors potentially accounting for lower vaccine uptake within ethnic minority populations (including lower trust resulting from poor healthcare experiences and awareness of past malpractice) might also be associated with lower rates of vaccine uptake among other groups, potentially including women, young people, less highly educated people and members of low-income households. On the other hand, one UK study has found that '[s]ocio-demographics do not explain vaccine hesitancy to any helpful degree' and that the major predictors of coronavirus vaccine hesitancy are 'excessive mistrust' and a lack of 'positive healthcare experiences'.[5] In context of the urgent need to understand and resolve disparities in vaccine coverage, using both quantitative and qualitative data as appropriate, these findings require interrogation through replication. For example, it might be that demographic variables only fail to explain variation following controls for trust and healthcare experiences because these arguably more proximal variables act to mediate the effects associated with certain demographic variables. Moreover, the use of more standard measures for trust and healthcare experiences would give greater confidence, as would preregistration of hypotheses.

This article reports on a preregistered study designed in order to test for demographic predictors of coronavirus vaccination uptake in the UK, both before and after controls for healthcare experiences and for trust in the principal institutions associated with the UK's coronavirus vaccination programme, that is, government, scientists and medical professionals.[25] As the UK's vaccination programme is delivered by National Health Service (NHS) bodies in England, Scotland, Wales and Northern Ireland, healthcare experience items focused specifically on the experiences of NHS healthcare. In order to provide the most robust test possible, established instruments have been used to measure both trust and healthcare experiences, rather than the novel measures employed in the study which found demographic variables not to be predictive.[5] Data collection and confirmatory analyses were pre-registered with the Center for Open Science (osf.io/56txk).

## Hypotheses

Given the observations regarding published findings of existing research, the following expectations were formed:

► H1. Coronavirus vaccination uptake will be lower among female respondents.
► H2. Coronavirus vaccination uptake will be lower among younger respondents.
► H3. Coronavirus vaccination uptake will be lower among respondents of other than white ethnic groups.

- ► H4. Coronavirus vaccination uptake will be lower among less educated respondents.
- ► H5. Coronavirus vaccination uptake will be lower among respondents from lower-income households.
- ► H6. Coronavirus vaccination uptake will be positively associated with trust in government, medical professionals and scientists.
- ► H7. Coronavirus vaccination uptake will be positively associated with positive experiences of healthcare.

## METHODS
### Study design
Trust was measured using Wellcome Global Health Monitor items Q11B, Q11E, Q14A and Q15a.[26] Experiences of healthcare were measured using items adapted from the Scottish Government Inpatient Experience Survey.[27] Responses to individual questions were numerically recoded and averaged to provide a single measure of trust and a single measure of experience of NHS healthcare. To facilitate analysis of individual trust and healthcare items, responses to these items were separately dichotomised for use in additional partial models, with the top two levels recoded as true and others recoded as false (this preregistered arbitrary cut-off point being chosen because it could be applied to all necessary variables regardless of how many levels they had).

Education was operationalised as highest qualification received, which was dichotomised in order to distinguish degree-level (and equivalent) qualifications (including undergraduate and postgraduate degrees) from lower levels of qualification, including lack of all formal qualifications. Ethnic group was measured using the categories employed in the 2011 UK census, which were subsequently dichotomised through coding as white or other than white, and household income was measured on an ordinal scale and dichotomised through coding as below median and median or above (for the sample). Dichotomisation of ethnicity has the unfortunate effect of homogenising the experiences of diverse minority groups, but was necessary because of the very small size of all minority groups when considered individually. Gender was measured by asking respondents whether they identified as 'male', 'female' or 'in another way'. As it was anticipated that relatively few respondents would not identify with one or other of the two canonical genders, gender was operationalised as a binary variable by reclassifying both 'male' and 'other' respondents as 'other than female'. This step proved to be unnecessary, as all participants in the sample identified either as male or as female. However, because it had been preregistered, the transformation was retained.

The dependent variable is a dichotomous variable representing the condition of either (a) having been vaccinated against the novel coronavirus (in the event that the respondent has been invited to be vaccinated) or of being about to attend an appointment to be vaccinated (in the event that a respondent was surveyed immediately after being invited to be vaccinated and for that reason has not been vaccinated yet) or (b) neither being vaccinated nor being about to be vaccinated despite having received an invitation.

All variables were measured through self-report as the questionnaire was completed online.

In calculating indices, missing values were ignored. Cases with missing data in relation to specific variables were excluded from all analyses featuring those variables.

### Data collection
Although there is no longitudinal aspect to the study reported here, the data collection formed part of a longitudinal study in which each wave involves collection both from recontactees and from new participants. Respondents were sampled at random from an Ipsos MORI recruited panel, with prestratification in order to produce representativeness of the UK adult population on age, gender, geographical region and working status. Where representativeness was not achieved, additional respondents were obtained by Ipsos MORI on a quota basis from panels maintained by other data providers. Following standard practice in the British polling industry, no quotas were applied with regard to ethnicity. However, overall proportions of white and other than white respondents within the sample were nonetheless broadly representative of the UK population (see following paragraph). Data were collected from respondents who have not been offered vaccination against the novel coronavirus, but data on these respondents were excluded from this particular study as they could not be used to test hypotheses about vaccine uptake.

Data collection was carried out online by Ipsos MORI. Fieldwork was launched on 1 April 2021 and completed on 16 April 2021. Informed consent was obtained from all participants. Demographic weights were calculated post collection by Ipsos MORI using the random iterative method on the basis of education and geographical region and of gender interlocked with age, National Readership Survey (NRS) social grade and working status. Weights were calculated before exclusion of cases where respondents had not yet been invited to receive coronavirus vaccination, as population margins for invited people specifically are not available. Data were collected from a total of 4885 respondents, of which 1662 were excluded due to not having received the invitation to be vaccinated. Table 1 provides a breakdown of the full sample by ethnicity, using the full range of UK census categories (note that it was possible to refuse consent for collection of data on ethnicity while giving consent to participate in the study as a whole). In comparison with the 2011 census, there was lower representation of the majority ethnic group, that is, white English/Welsh/Scottish/Northern Irish/British, than in the 2011 census, along with greater representation of individuals of other white background, although these discrepancies might plausibly be attributed to actual population changes in the intervening decade (findings of the 2021 census have not yet been released); other groups are so small

**Table 1** Ethnic groups in the full sample

| Ethnic group | n | % |
|---|---|---|
| White English/Welsh/Scottish/Northern Irish/British | 4009 | 82 |
| White Irish | 49 | 1 |
| White Gypsy or Irish Traveller | 1 | 0 |
| Any other white background | 319 | 7 |
| White and black Caribbean | 22 | 0 |
| White and black African | 12 | 0 |
| White and Asian | 32 | 1 |
| Any other mixed/multiple ethnic background | 22 | 0 |
| Indian | 78 | 2 |
| Pakistani | 77 | 2 |
| Bangladeshi | 39 | 1 |
| Chinese | 40 | 1 |
| Any other Asian background | 22 | 0 |
| African | 44 | 1 |
| Caribbean | 33 | 1 |
| Any other black/African/Caribbean background | 9 | 0 |
| Arab | 9 | 0 |
| Any other ethnic group | 14 | 0 |
| Prefer not to answer | 42 | 1 |
| Consent not granted | 12 | 0 |

that differences in representation are probably not meaningful to discuss (eg, 2% Indian in this sample vs 2.5% in the 2011 census). The number of included cases with missing values for each variable is listed in table 2, with a breakdown by dichotomised ethnicity. This shows that there was a higher rate of missing data for NHS healthcare among white respondents than among other than white respondents, amounting to roughly 3% of included cases in the former case (in large part because the individuals concerned did not use NHS healthcare services), and a higher rate of missing data for vaccine uptake among other than white respondents than among white respondents, amounting to roughly 1% of included cases in the former case.

### Statistical analysis

The sample was treated as equivalent to a random sample for analytic purposes. Hypotheses 1–7 were tested through creation of binomial logit models with vaccine acceptance as the dependent variable. Of most relevant interest are a partial model using only demographic predictors and the full model featuring demographic predictors and aggregate measures of trust and of experiences of NHS healthcare. Both of these models were preregistered; as explained in the preregistration document, the full model was considered the definitive test of the hypotheses, with the other models provided for information. To further probe the findings of the confirmatory analysis (see below), two further partial models were created without having been preregistered. These included one featuring demographic predictors and aggregate measures of trust only and one featuring demographic predictors and aggregate measures of NHS healthcare only. In order to study the predictiveness of individual items, two further additional preregistered models were created, one featuring demographic predictors and dichotomised responses to trust items as predictors, and the other featuring demographic variables and dichotomised responses to healthcare experience items as predictors, although (as noted above) it was the full model that was considered the definitive test. To further understand the unexpected findings of the confirmatory analyses, mediation analyses were conducted on an exploratory basis using non-parametric bootstrapping with 10 000 repetitions in order to test for theoretically plausible mediatory relationships suggested by the finding that certain variables were significantly predictive in partial models but not in the full model. Being developed in response to the findings of the preregistered analyses, these mediation analyses are detailed below. (The unusually high number of repetitions was chosen in order to reduce random variation, despite the increase in demands on computer processing power.) Coefficients for all confirmatory and exploratory analyses are reported both as estimates and as 95% CIs.

### Power analysis

Power analysis confirms that the sample was sufficiently large to detect even very small effects, at least in the preregistered models. Given a sample size of 2773 (ie, the number of observations which were sufficiently complete to be fitted in the full model) and a cut-off of $p<0.050$ (two-tailed), an association so weak as to reduce the probability of vaccine uptake only from 95% to 91% across the entire range of a predictor variable could be detected with approximately 98% power.

**Table 2** Included cases with missing values

| | Age | Gender | Degree | Other than white | Low income | Trust | NHS experiences | Uptake |
|---|---|---|---|---|---|---|---|---|
| Overall | 0 | 0 | 0 | 22 | 305 | 8 | 124 | 46 |
| White | 0 | 0 | 0 | | 267 | 5 | 121 | 27 |
| Other than white | 0 | 0 | 0 | | 26 | 1 | 2 | 14 |

NHS, National Health Service.

**Table 3**  Descriptive statistics: demographic variables

| | Age | | | Gender | | Degree | | Ethnic group* | | Household income† | | |
|---|---|---|---|---|---|---|---|---|---|---|---|---|
| | **n** | **M** | **SD** | **Female (%)** | **Male (%)** | **Yes (%)** | **No (%)** | **White (%)** | **Other than white (%)** | **Low (%)** | **Middle (%)** | **High (%)** |
| **All invited** | | | | | | | | | | | | |
| Raw | 3223 | 52.00 | 14.64 | 51 | 49 | 39 | 61 | 93 | 7 | 43 | 19 | 38 |
| Weighted | | 51.89 | 14.70 | 51 | 49 | 28 | 72 | 92 | 8 | 44 | 20 | 37 |
| **Uptake: yes** | | | | | | | | | | | | |
| Raw | 2990 | 52.78 | 14.40 | 51 | 49 | 40 | 60 | 94 | 6 | 43 | 19 | 38 |
| Weighted | | 52.70 | 14.48 | 51 | 49 | 28 | 72 | 93 | 7 | 44 | 19 | 37 |
| **Uptake: no** | | | | | | | | | | | | |
| Raw | 187 | 44.10 | 13.83 | 50 | 50 | 34 | 66 | 85 | 15 | 42 | 23 | 35 |
| Weighted | | 44.01 | 13.65 | 49 | 51 | 24 | 76 | 84 | 16 | 42 | 24 | 33 |

*White ethnic groups comprise white British, English, Welsh, Scottish, Northern Irish, Irish, Gypsy or Irish Traveller, and any other white background; other than white ethnic groups comprise all other ethnic categories used in the 2011 UK census, including mixed ethnicities.
†Low: £24 999 or less; middle (median for included participants): £25 000–£34 999; high: £35 000 or more.

## Patient and public involvement

No patients were involved.

## RESULTS

### Descriptive statistics and internal reliability

Cronbach's alpha and Guttman's lambda 6 were used to assess internal reliability of indices. The reliability of items used to measure experiences of NHS healthcare was assessed as $\alpha=0.88$ and $\lambda_6=0.87$, which indicate a high level of internal reliability. The reliability of items used to measure trust was assessed as $\alpha=0.69$ and $\lambda_6=0.66$, which would be a suboptimal level of internal reliability for a conventional Likert scale but likely reflects real differences in the level of trust for different institutions (especially given that government trust may be acting at least partly as a proxy for political preference).

Table 3 shows the weighted and unweighted demographic descriptive statistics, first for all included respondents and then for respondents who did and did not accept the invitation to be vaccinated against the novel coronavirus. Although respondents from households with income at or above the median for the sample were treated as a single group for modelling purposes, they are disaggregated here. It will be noted that the percentage of white respondents is higher than in the raw sample. However, this was to be expected as inclusion was limited to individuals who had been offered vaccination in a context where vaccination had been offered in descending order of age and where other than white ethnic groups have a younger age profile.[28] Table 4 shows that this national trend could also be observed among respondents in the current study. It is acknowledged that the number of other than white respondents was small in absolute terms, but it is noted that collection of a larger number would have required deliberate oversampling (and therefore a less representative sample).

The mean age for respondents who had been invited to be vaccinated yet had not taken up the invitation is just under 9 years younger than the mean age for respondents who had been vaccinated. Percentages of male and female respondents in each group are effectively identical. Percentages with low (below median) household income for the sample are effectively identical, but the weighted percentage of respondents with high (above median) household income for the sample is 4 points lower among those who did not accept the invitation to be vaccinated (ie, the proportion was about 0.9 times as high). This suggests that the effect of income may be masked in the confirmatory analysis presented in this article, as dichotomisation by separating low-income households from median-income and high-income households was a preregistered data transformation: a point to which we shall return in the Results section. The weighted percentage of respondents educated to degree level was 4 points lower among those who did not accept the invitation to be vaccinated (ie, it was about 0.9 times as high). The weighted percentage of respondents belonging to other than white ethnic groups was 9 points higher among those who did not accept the invitation to be vaccinated (ie, it was about 2.3 times as high). This is not a large difference: by the time of data collection,

**Table 4**  Age profile of dichotomised ethnic groups among the included respondents

| Ethnicity | n | M | SD |
|---|---|---|---|
| White | 2967 | 53.14 | 14.12 |
| Other than white | 234 | 38.15 | 13.91 |

**Table 5** Descriptive statistics: indices

| | | NHS experiences | | Trust | |
|---|---|---|---|---|---|
| | **n** | **M** | **SD** | **M** | **SD** |
| All invited | | | | | |
| Raw | 3223 | 4.06 | 0.71 | 3.09 | 0.52 |
| Weighted | | 4.05 | 0.71 | 3.08 | 0.53 |
| Uptake: yes | | | | | |
| Raw | 2990 | 4.08 | 0.69 | 3.12 | 0.48 |
| Weighted | | 4.08 | 0.69 | 3.12 | 0.48 |
| Uptake: no | | | | | |
| Raw | 187 | 3.70 | 0.92 | 2.64 | 0.77 |
| Weighted | | 3.69 | 0.94 | 2.64 | 0.78 |

NHS, National Health Service.

94% and 87% of invited members of white and other than white ethnic groups within the sample who had received the invitation to be vaccinated had taken it up, which imply that the great majority of the members of all ethnic groups had done so. However, it is consistent with the observation, discussed above, that uptake rates have been lower among members of other than white ethnic groups.

Table 5 shows the weighted and unweighted indices of trust and NHS healthcare experiences, first for the whole sample and then for respondents who did and did not accept the invitation to be vaccinated against the novel coronavirus. The mean scores for members of the latter group were lower on both indices. Table 6 shows the unweighted product-moment correlations between all pairs of variables. There was a strong correlation between indices of trust and NHS healthcare experiences (r=0.47), but NHS healthcare experiences were less strongly correlated with vaccine uptake (r=0.13) than trust was (r=0.22), while trust was less strongly correlated with membership of an other than white ethnic group (r=−0.08) than NHS healthcare experiences were (r=−0.13). Age was negatively correlated with membership of an other than white ethnic group (r=−0.27) and with degree-level education (r=−0.09), but positively

correlated with membership of a low-income household (r=0.11) and with vaccination uptake (r=0.14).

## Confirmatory analysis

Table 7 presents the adjusted OR (AOR) with 95% CI for dichotomous variables (female gender, membership of an other than white ethnic group, membership of a low-income household, and education to undergraduate or postgraduate degree level) and for increases of 1 SD in numeric variables (ie, age and the indices for trust and NHS healthcare experiences). Age (AOR=1.61, 95% CI 1.39 to 1.87, p<0.001) and membership of an other than white ethnic group (AOR=0.53, 95% CI 0.35 to 0.84, p=0.005) were the only statistically significant predictors in the purely demographic model. Thus, the null hypothesis can be rejected with regard to H2 and H3, but not with regard to H1, H4 and H5.

Controlling for indices of trust and of NHS healthcare experiences diminishes the effect associated with membership of an other than white ethnic group from AOR=0.53 (95% CI 0.35 to 0.84, p=0.005) to AOR=0.61 (95% CI 0.38 to 1.01, p=0.046). This suggests that some of the effect associated with ethnicity may be mediated by one or both of these additional variables. However, the same cannot be said for age. Indeed, the null hypothesis can only be rejected for H3 with regard to this model at p<0.050, but the coefficient and p value relevant to H2 are virtually unchanged relative to the demographic-only model. Moreover, while there was a significant effect associated with trust in the full model (AOR=2.02, 95% CI 1.72 to 2.37, p<0.001), there was no effect associated with NHS healthcare experiences (AOR=1.02, 95% CI 0.86 to 1.2, p=0.820). Thus, the null hypothesis can be rejected with regard to H6, but not with regard to H7.

This finding would appear to be in contradiction to the published claim that vaccine hesitancy is largely explained by trust alongside healthcare experiences and that demographic predictors have little importance. Partial models excluding trust or healthcare experiences are also presented in table 7. As the effect associated with NHS healthcare experiences in the partial model excluding trust is very highly significant (AOR=1.46, 95% CI 1.27 to

**Table 6** Key variables: bivariate correlations

| | 1 | 2 | 3 | 4 | 5 | 6 | 7 | 8 |
|---|---|---|---|---|---|---|---|---|
| 1. Age | | −0.01 | −0.27 | 0.11 | −0.09 | 0.10 | 0.06 | 0.14 |
| 2. Female | −0.01 | | −0.04 | 0.05 | 0.05 | 0.01 | 0.05 | 0.00 |
| 3. Other than white | −0.27 | −0.04 | | −0.01 | 0.03 | −0.13 | −0.08 | −0.08 |
| 4. Low income | 0.11 | 0.05 | −0.01 | | −0.21 | −0.03 | −0.06 | 0.00 |
| 5. Degree | −0.09 | 0.05 | 0.03 | −0.21 | | 0.04 | 0.04 | 0.03 |
| 6. NHS experiences | 0.10 | 0.01 | −0.13 | −0.03 | 0.04 | | 0.47 | 0.13 |
| 7. Trust | 0.06 | 0.05 | −0.08 | −0.06 | 0.04 | 0.47 | | 0.22 |
| 8. Uptake | 0.14 | 0.00 | −0.08 | 0.00 | 0.03 | 0.13 | 0.22 | |

NHS, National Health Service.

**Table 7** Partial models and full model

| | AOR | 2.5% | 97.5% | P value |
|---|---|---|---|---|
| Demographics only | | | | |
| Age | 1.61 | 1.39 | 1.87 | <0.001 |
| Female | 0.96 | 0.71 | 1.30 | 0.808 |
| Other than white | 0.53 | 0.35 | 0.84 | 0.005 |
| Low income | 1.02 | 0.75 | 1.40 | 0.883 |
| Degree | 1.30 | 0.92 | 1.87 | 0.140 |
| NHS experiences | | | | |
| Age | 1.56 | 1.34 | 1.83 | <0.001 |
| Female | 0.98 | 0.71 | 1.34 | 0.882 |
| Other than white | 0.65 | 0.41 | 1.04 | 0.064 |
| Low income | 1.09 | 0.79 | 1.51 | 0.591 |
| Degree | 1.29 | 0.91 | 1.87 | 0.167 |
| NHS experiences | 1.46 | 1.27 | 1.67 | <0.001 |
| Trust | | | | |
| Age | 1.57 | 1.34 | 1.85 | <0.001 |
| Female | 0.89 | 0.65 | 1.23 | 0.490 |
| Other than white | 0.61 | 0.39 | 0.98 | 0.037 |
| Low income | 1.12 | 0.81 | 1.56 | 0.491 |
| Degree | 1.18 | 0.82 | 1.71 | 0.380 |
| Trust | 2.06 | 1.80 | 2.36 | <0.001 |
| Full model | | | | |
| Age | 1.57 | 1.33 | 1.86 | <0.001 |
| Female | 0.92 | 0.66 | 1.28 | 0.615 |
| Other than white | 0.61 | 0.38 | 1.01 | 0.046 |
| Low income | 1.16 | 0.83 | 1.63 | 0.379 |
| Degree | 1.22 | 0.84 | 1.80 | 0.297 |
| NHS experiences | 1.02 | 0.86 | 1.20 | 0.820 |
| Trust | 2.02 | 1.72 | 2.37 | <0.001 |

AOR, adjusted OR; NHS, National Health Service.

**Table 8** Individual trust and NHS healthcare experience items (dichotomised)

| | AOR | 2.5% | 97.5% | P value |
|---|---|---|---|---|
| NHS experiences | | | | |
| Age | 1.51 | 1.30 | 1.77 | <0.001 |
| Female | 0.91 | 0.67 | 1.24 | 0.556 |
| Other than white | 0.64 | 0.41 | 1.01 | 0.049 |
| Low income | 1.03 | 0.75 | 1.41 | 0.848 |
| Degree | 1.24 | 0.88 | 1.79 | 0.226 |
| Respect | 0.99 | 0.61 | 1.59 | 0.967 |
| Care | 1.17 | 0.72 | 1.89 | 0.513 |
| Needs | 1.45 | 0.96 | 2.18 | 0.077 |
| Trust staff | 1.62 | 1.01 | 2.58 | 0.045 |
| Understand | 1.37 | 0.89 | 2.11 | 0.148 |
| Spiritual | 0.76 | 0.54 | 1.07 | 0.113 |
| Trust | | | | |
| Age | 1.52 | 1.30 | 1.78 | <0.001 |
| Female | 0.91 | 0.67 | 1.24 | 0.552 |
| Other than white | 0.68 | 0.43 | 1.09 | 0.097 |
| Low income | 1.12 | 0.82 | 1.54 | 0.487 |
| Degree | 1.20 | 0.84 | 1.72 | 0.322 |
| Trust UK government | 1.48 | 1.07 | 2.07 | 0.021 |
| Trust scientists working at universities in the UK | 1.77 | 1.12 | 2.77 | 0.013 |
| Trust scientists working at private companies in the UK | 1.94 | 1.34 | 2.78 | <0.001 |
| Trust medics | 2.81 | 1.74 | 4.49 | <0.001 |

AOR, adjusted OR; NHS, National Health Service.

1.67, p<0.001), although lower than the effect associated with trust in the partial model excluding NHS healthcare experiences (AOR=2.06, 95% CI 1.8 to 2.36, p<0.001; AOR=2.06, 95% CI 1.8 to 2.36, p<0.001), it seems plausible that the effect of NHS healthcare experiences on uptake may be mediated by trust: a possibility to which we shall return in the exploratory analysis presented in the following subsection.

Table 8 presents the AOR with 95% CI for increases of 1 SD in age and for dichotomous variables, including dichotomised trust and NHS healthcare items as individual predictors. Examination of coefficients associated with these items facilitates assessment of their individual predictive contribution. All effects associated with trust items are statistically significant, especially those associated with trust in medical professionals (AOR=2.81, 95% CI 1.74 to 4.49, p<0.001) and with scientists working in the private sector (AOR=1.94, 95% CI 1.34 to 2.78,

p<0.001). Thus, high trust in any of the institutions mentioned was associated with increased likelihood of uptake, even after controlling for level of trust in all the others, plus demographic variables. Interestingly, the only significant effect associated with an individual NHS healthcare experience item after controlling for all the other items and for demographic variables was that associated with having felt able to trust the staff (AOR=1.62, 95% CI 1.01 to 2.58, p=0.045). In relation to these models, the null hypothesis is still rejected with regard to H6 and may be partially rejected with regard to H7.

### Exploratory analysis

The above findings were suggestive of theoretically plausible mediated associations, that is, that experiences of NHS healthcare might be indirectly associated with uptake via trust, that ethnicity might be indirectly associated with trust via experiences of NHS healthcare, and

that ethnicity might therefore be indirectly associated with uptake via both of the aforementioned. For this reason, additional mediation analyses were carried out on an exploratory basis (ie, without having been preregistered). All of these analyses involved the same demographic controls used in the confirmatory analyses.

There were three findings, here reported in relation to linear models with standardised predictor and outcome variables. First, the relationship between membership of an other than white ethnic group and trust appears to be 100% (95% CI 48% to 100%, p=0.040) mediated by NHS healthcare experiences. Second, the relationship between NHS healthcare experiences and uptake appears to be 94% (95% CI 56% to 100%, p<0.001) mediated via trust. Third, the association between membership of an other than white ethnic group and uptake appears to be 24% (95% CI 8% to 100%, p=0.024) mediated via NHS healthcare experiences (trust was not included in this model, given the second finding). This mediating effect may plausibly be taken to account for the reduction in the effect associated with ethnicity between the demographic-only model and the full confirmatory model.

While these findings are exploratory rather than confirmatory, they add weight to the above interpretation of the confirmatory findings. On the one hand, they suggest that positive experiences of NHS healthcare may be associated with higher vaccine uptake largely for the reason that they have a positive relationship with trust in relevant institutions, and that membership of an other than white ethnic group is associated with lower trust in those same institutions for the reason that experiences of NHS healthcare are less positive among members of such groups. On the other hand, the findings also suggest that these intermediary relationships can only account for some of the association between ethnicity and vaccine uptake. Most of the associations remain unexplained, as does the association between age and uptake.

## DISCUSSION

The finding that female gender, below degree-level education and below median income do not predict uptake contradicts several studies which measured vaccine hesitancy before the UK vaccination campaign had got underway.[6–8 10] This discrepancy may be interpreted in terms of an attitude change potentially attributable to public health communication, word-of-mouth discussion of the coronavirus vaccination itself or many other sources (although it is also possible that members of certain groups simply underestimated their own likelihood of accepting vaccination).

The finding that age and ethnicity do, however, predict uptake even after controls for trust and NHS healthcare experiences is in contradiction to the earlier finding that vaccine hesitancy is largely explained by mistrust and poor experiences of healthcare, to such an extent that demographic variables are not predictive once those more proximal variables are controlled for.[5] Non-replication of that

finding might be accounted for by the timeframe (data collected after the vaccination programme had begun), outcome measure (actual uptake rather than expected uptake), predictor measures (those used here both for trust and for healthcare experiences were more standard) or analytic approach (preregistered models rather than exploratory modelling). Age is here found to remain a powerful predictor of uptake even after those controls and ethnicity is also found to predict uptake, although this effect appears weaker after the same controls. The finding that age is positively associated with uptake replicates an earlier large cohort study[10] and is consistent with several studies which have found a negative association between age and coronavirus vaccine hesitancy.[6–8] The reduction in strength of association for ethnicity following controls appears to be the result of mediation by NHS healthcare experiences, via trust in the institutions involved in developing and delivering the vaccine, which provides support for the argument that lower vaccine uptake in ethnic minority populations might be driven via lower trust resulting from worse healthcare experiences.[22] However, the mediation effect is small, appearing to account only for about a quarter of the variation in uptake which was found to be associated with ethnicity in the demographic-only model. To summarise, this study finds evidence that uptake of coronavirus vaccination is lower among members of other than white ethnic groups, that NHS healthcare experiences have a relationship with vaccine uptake that is mediated by trust, and that the lower trust in medics, scientists and government among members of other than white ethnic groups is mediated by their worse experiences of NHS healthcare. However, it does not find evidence that worse healthcare experiences or lower trust can explain more than a small part of the ethnicity gap in coronavirus vaccine uptake.

This finding (or non-finding) suggests that lower rates of vaccination uptake among members of other than white ethnic groups are likely to have drivers other than or in addition to mistrust in medical, scientific and other authorities and to negative experiences of healthcare. Experiences of healthcare explain little or no variation in uptake beyond that which is explained by trust, their association with uptake being almost entirely mediated by the latter. In some ways, this is encouraging: had the effect associated with ethnicity been found to be entirely explicable in terms of worse experiences of healthcare, then prospects of reducing it in the short term—and especially under pandemic conditions—would appear very bleak. However, the fact that so much variance remains unexplained is in itself a cause for concern and highlights the urgent need for further research to test the mediating role of other variables, alongside qualitative research aiming to uncover the detailed mechanisms by which ethnicity and other demographic variables might be related to uptake. Possible candidate mediators involve access barriers with regard to coronavirus vaccination services,[22] as well as sources of information, conspiracy beliefs about the novel coronavirus and attitudes towards vaccination

in general.[8] Moreover, the finding that ethnic disparities in trust—a vital component in public health—could be so closely related to poorer reported experiences of NHS healthcare should serve as a reminder of the very real problem of ethnic disparities in healthcare experiences and outcomes, whether those disparities can be assumed to have a population-level impact on coronavirus vaccine uptake or not.

## Limitations of this study

This study relies on self-report data collected and as such it is dependent both on respondents' candour and on their recall. Also, while much effort has been made to produce a representative sample, it was not possible to obtain a probability sample. Moreover, as noted above, the focus on low income as a predictor (to which the researchers were committed by the preregistration document) may have concealed the importance of high income as a predictor. In addition, the data collected here provide no basis on which to ascertain whether the effect associated with age reflects a genuine tendency towards lower uptake or rather results simply from the UK's vaccination schedule: because older people were invited first, they will have had more time in which to be vaccinated (although as noted above, other studies have found a negative association between age and vaccine hesitancy). Lastly, while the researchers were working with a relatively large sample, it included only 187 individuals who did not take up the invitation to be vaccinated and only 234 members of other than white ethnic groups. Given that any effects associated with ethnicity must be disentangled from those associated with age, which in this context functions as a confound (both because age had a stronger association with uptake than ethnicity and because members of other than white ethnic groups were on average younger than members of white ethnic groups in the sample, as in the wider UK population), special caution must therefore be exercised in interpreting the results presented here. Further study of relationships such as those focused on in this study, for example looking for variables that might mediate the relationship between age and uptake or that might explain the currently unexplained majority of the relationship between ethnicity and uptake, might therefore benefit either from an even larger sample or from deliberate oversampling of individuals who have not taken up the invitation to be vaccinated and perhaps also of members of specific demographic groups.

## Technical note

The analysis presented here was carried out using R V.4.0.5,[29] with use of the following packages: psych V.2.1.3[30] for calculation of internal reliability of scales, Hmisc V.4.5.0[31] for calculation of weighted SD, BBmisc V.1.11[32] for standardisation of numeric variables, mediation V.4.5.0[33] for mediation analysis and WebPower V.0.6[34] for power analysis.

**Contributors** DA, SM, BD and VM-H were jointly responsible for the design of the data collection instrument. DA planned, preregistered and executed the analysis presented here, and wrote this article together with SM, who has overall leadership of the project within which this research took place. Revisions of the article following peer review were implemented by DA, who is also responsible for the overall content as the guarantor.

**Funding** This research was supported by the Economic and Social Research Council (ES/V 015494/1).

**Competing interests** None declared.

**Patient and public involvement** Patients and/or the public were not involved in the design, or conduct, or reporting, or dissemination plans of this research.

**Patient consent for publication** Not required.

**Ethics approval** This study involves human participants and was approved by King's College London Ethics Committee (MRA-20/21-21420). Participants gave informed consent to participate in the study before taking part.

**Provenance and peer review** Not commissioned; externally peer reviewed.

**Data availability statement** Data are available upon reasonable request. Data will be made available following the conclusion of the project's period of funding.

**ORCID iD**
Daniel Allington http://orcid.org/0000-0002-5378-0552

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
