## [Reviewer comments · BMJ Open]

ARTICLE DETAILS

TITLE (PROVISIONAL)	Trust and experiences of National Health Service healthcare do not fully explain demographic disparities in coronavirus vaccination uptake in the UK: a cross-sectional study
AUTHORS	Allington, Daniel; McAndrew, Siobhan; Duffy, Bobby; Moxham-Hall, Vivienne

VERSION 1 – REVIEW

REVIEWER	Mosser, Jonathan University of Washington, Health Metrics Sciences
REVIEW RETURNED	01-Aug-2021

GENERAL COMMENTS	In this manuscript, the authors analyze data from a cross-sectional survey in April 2021 of UK residents, investigating associations between reported ethnicity, trust in government, scientists, and medical professionals, and SARS-CoV-2 vaccine uptake. The article takes up an important question: what are the drivers of gaps in coronavirus vaccine uptake by race/ethnicity in the United Kingdom? (There are also examinations of other demographic associations, but the article focuses primarily on race/ethnicity). With that said, I have a number of comments about the manuscript, which are detailed below. In general, I think that the authors need to discuss more broadly the various mechanisms by which the observed disparities – particularly between racial / ethnic groups – may arise, beyond those examined here (i.e. via healthcare experiences and trust). There are several study limitations that I think should be mentioned or discussed in more detail. Last, while the methods generally appear to be appropriate, there are opportunities to increase the clarity of the writing so that readers can better understand what exactly these data and analyses can – and can't – tell us. Last, as I have commented to the editor, I would acknowledge my limited expertise as a reviewer in several critical areas for this study. I am not an expert in racism and its pathophysiologic mechanisms as relate to disparities in vaccination coverage; as a white male from the United States, I also do not have expertise in the particular drivers of racial/ethnic or other inequities in vaccination coverage in the UK. I urge that reviewers with this expertise be assigned to this manuscript as well. In the absence of personal expertise, I would note that my comments below draw largely from Boyd, Lindo, Weeks, and McLemore's framework (https://www.healthaffairs.org/doi/10.1377/hblog20200630.939347/full/), which develops standards for publishing on racial health inequities. I would defer to reviewers with more specific expertise in these areas, however, if their comments should conflict with mine.
--

Major comments:

MAJOR COMMENT #1

The authors focus primarily on the connections between vaccine hesitancy and vaccine uptake in this manuscript. For example, in page 5 / lines 4-9, a direct line is drawn between vaccine confidence and uptake: “However, despite this encouraging trend, demographic disparities appeared to persist [11], albeit at a reduced level, with youth and membership of other than white ethnic groups now being the strongest predictors of coronavirus vaccine hesitancy in the UK [10], such that by 13 January 2021, 42.5% of white people aged over 80 but not resident in care homes, but only 20.5% of black people in the same category, had been vaccinated”.

Of course, vaccine confidence is only one of the factors that affect vaccination uptake. For instance, Razai, Osama, and McKechnie describe concerns about access barriers, including locations of vaccine delivery, time of availability, residential segregation as a form of systemic racism (BMJ 2021;372:n513). Access barriers are also cited by the UK SAGE ethnicity sub-group (“Factors influencing COVID-19 vaccine uptake among minority ethnic groups, 17 December 2020”, <https://www.gov.uk/government/publications/factors-influencing-covid-19-vaccine-uptake-among-minority-ethnic-groups-17-december-2020>).

I appreciate that the analyses presented in this article focus on the associations between ethnicity, trust, NHS experiences, and vaccine uptake. I think that it is important, however, to more broadly consider the range of potential drivers for such disparities when discussing these findings. For instance, the authors do mention in the introduction (i.e. page 5, lines 23-27) that racism, discrimination, and exclusion have the potential to contribute to vaccine hesitancy and circulation of misinformation, and through this mechanism be associated with decreased vaccine uptake. Beyond this one mechanism, however, I would strongly suggest that the authors engage more broadly – in both the introduction and discussion - with the variety of potential ways that historical and present racism may contribute to differential vaccination rates (for COVID and non-COVID vaccines) or access to similar health services.

Along the same lines, on page 15 the authors write, “However, the fact that so much variance remains unexplained is in itself a cause for concern, and highlights the urgent need for further research to test the mediating role of other variables. Possible candidate mediators involve sources of information, conspiracy beliefs about the novel coronavirus, and attitudes to vaccination in general”. First, I might suggest that the authors highlight the urgent (broader) need to understand and resolve disparities in vaccination coverage, rather than focusing so narrowly on the need for research to test various potential mediating variables. While statistical analyses like these are useful, during a rapidly evolving vaccine roll-out programs may require the use of more qualitative information, for instance, to respond in real-time to emerging disparities. Second, the authors highlight some potential reasons why their models can only explain some of the relationship between uptake and ethnicity. While “conspiracy beliefs” or attitudes towards vaccination in general may play a role, I think that it is important to contextualize these more fully. As above, might differential access play a role in these gaps? What factors might lead members of other-than-white ethnic groups to have different attitudes towards vaccination in general and/or hesitancy about a new vaccine (systemic racism and discrimination, legacies of previous unethical research,

underrepresentation of racial/ethnic groups in medical research, whether as participants or researchers, etc.)?)

MAJOR COMMENT #2:

I have several related comments regarding the classification of ethnic group used in this study.

First, please include additional detail about how ethnic group was ascertained. Was this via self-report? If so, how was the question asked (as this may presumably influence the nature of the responses)? Was a post-hoc classification used (the footnote in Table 4.1 seems to indicate as much)? If so, how was this post-hoc grouping determined – by the authors, or was another standard classification applied? Were individuals allowed to choose more than one racial/ethnic category? If so, how were those individuals classified in this study?

Second, the authors note that comparison with official statistics suggests “slight representation of white ethnic groups” (pg 9, line ~ 38 – is this a typographical error for “overrepresentation”). It would be useful to have more detail to judge how representative this sample is (or is not) with respect with ethnicity within the UK. For instance, pg 7 / lines 6-10 seems to suggest that pre-stratification was used to produce representativeness of the UK adult population by age, gender, geographic region, and working status – but not by ethnicity. Is that true? If there was no pre-stratification or other effort in the sampling design to ensure that the sample was representative by ethnicity, are there any other data that would help the reader judge whether or not this population was likely to be representative by ethnicity? For instance, is there any information available on non-response rates by ethnicity? Were there any trends in the missingness mentioned in table 3.1 (by ethnicity or otherwise), or was this missingness random?

Related: the authors on page 5 lines 29-30 write that one of the strengths of the study was “representativeness of the UK adult population” - but that conflicts with the limitation in the third bullet point about limited representation of members of other-than-white ethnic groups. It seems that this statement about representativeness of the UK adult population should be revised in this light.

Third, the analysis dichotomizes ethnic groups into “white” or “other than white”. I appreciate that this may have been a practical measure necessitated by sample size. However, this grouping homogenizes the diverse experiences of different other-than-white ethnic groups in the UK. At a minimum, this should be discussed as an important limitation of the study. In addition, it would be helpful if the authors could include information – if available - on which “other than white” ethnic groups were represented in their data set and if this distribution is reflective of the general UK population.

MAJOR COMMENT #3:

As the authors note, one of the substantial limitations of this analysis (and the underlying data) is the potential for confounding between age and identification as a member of an other-than-white ethnic group (bivariate correlation of $-.27$ in table 4.3), both of which were associated with vaccine uptake, NHS experiences, and trust.

In addition, it seems like a significant limitation that – at the time of the survey – the vaccine had been available to older age groups for a

longer period of time than it had for younger age groups. The association between age and uptake may therefore represent – at least in part - an artifact of the longer time of availability at the time of the survey. I would encourage the authors to add this as a potential limitation to the analysis, if thought to be applicable giving the timing of vaccination rollout by age and this data collection.

As the authors note, multiple logistic regression cannot fully eliminate the potential impact of this sort of a confounder. Did the authors consider any other sensitivity analyses to further investigate the interaction between age and ethnicity? It may be helpful to at least provide some additional description of the differences in age between ethnic groups. For instance, a negative correlation between age and membership in an other-than-white ethnic group. It would be useful, however, to report more clearly what the distribution of age was in respondents who identified as white vs those who identified as other-than-white (ideally contextualizing this with respect to which age groups had access to the vaccine at which time). If the distributions are largely similar, at least with respect to COVID risk and vaccine availability by age, then the confounding may be less of an issue. If members of other-than-white ethnic groups were substantially younger, and the younger surveyed individuals in these groups tended to have lower vaccine uptake, then it is harder to interpret the findings of this study.

Last, my impression is that this is one of the major limitations of the study and should be included in the “strengths and limitations” bullet points on page 5.

MAJOR COMMENT #4:

I think that there are some opportunities to clarify the methodological approach and results. After several readings of the article, I think that the methods and mediation analyses were largely appropriate, but it was a little difficult to understand at first exactly what the authors had done.

First, I would suggest expanding the methods section – especially section 3.3, “Statistical Analysis”. In particular, more detail about exactly what mediation analyses were performed and why would be useful. Some of these details are in the results section, but in general I was not sure which mediation analyses were performed, how they were selected, and what specific methods were used.

Second, I appreciate that the authors clearly lay out their hypotheses. In reading the results, however, it is easy for the reader to get lost in the terminology (H1 vs H4, etc.) and the “rejection of the null hypothesis” phrasing. I found myself often having to refer back to the numbered list of hypotheses to understand what the writing meant. I would encourage the authors to more clearly state their findings in plain rather than statistical language and refer to hypothesis numbers sparingly if at all. It is also confusing to see that, in some cases, full models and in other cases partial models are used to evaluate the various hypotheses. For instance, on page 13, the authors state (using a full model) “Thus, the null hypothesis can be rejected with regard to H6, but not with regard to H7.” But the partial model including just NHS experiences and demographic predictors shows a significant effect for NHS experiences. And, earlier in the section (lines 45-46, page 12) the authors test a number of hypotheses using the demographics-only model. Essentially, there are many models being run, and it is unclear

	to me how the authors are choosing which model to use to test which hypotheses in some cases. Minor comments: #1: I am not sure what the sentence on page 5 lines 37-47 means: “Moreover, it would appear plausible that the range of mechanisms which can be assumed to generate the forms of exclusion associated with racism might also be associated with decreased trust, more negative experiences of healthcare, and higher rates of vaccine hesitancy among other groups, including women, young people, less highly educated people, and members of low income households”. In particular, I am not sure what the “range of mechanisms which can be assumed to generate the forms of exclusion associated with racism” is meant to represent in this context. Is there a way to phrase this more clearly? #2: Page 7 lines 42-44: I'm a bit confused by how gender was ascertained. It seems here that survey participants were asked to categorized themselves as “female” or “other than female”. However, the text goes on to say that all individuals identified as either male or female. How was this information ascertained? Perhaps I'm missing something – did the actual survey include more gender groupings (or free response, etc) and then the authors re-categorized into female and other than female? #3: Table 4.1: Are some of these values in the table percentages? If so, would clarify either in the cells themselves or in the headers. Also, please ensure that all abbreviations used in table headers are defined. #4: Table 4.1 and 4.2; page 9 / line 33: Please explain in the methods how the weights used for these weighted analyses were derived. #5: Page 9 line 39: should “representation” be “overrepresentation” here?
--	--

REVIEWER	Tabassum, Mimma Noakhali Science and Technology University, Statistics
REVIEW RETURNED	25-Aug-2021

GENERAL COMMENTS	The manuscript is very convenient for the present condition. I am appreciated for their work but I have some queries about the manuscript. in the title (page 3), would you please kindly include the full form of NHS? in the section study design (page 6), in line 35, would you please kindly explain how did you choose top two levels recorded as true and other recorded as false? in the section statistical analysis (page 7), in line 50, would you please kindly explain why are you taking 10,000 repetitions? in the result section (page 8), in table 4.1, would you please kindly explain all the numbers are in percentage? in the power analysis section (page 8), in line 1, would you please kindly explain how you find the sample size of 2773 for power calculation?
--

	in the result section (page 9), in table 4.3, would you please kindly explain how do you select the variables for bivariate correlations? in the discussion section (page 14), you discussed only the result of your study but there is no comparison with the published study's result in UK or other countries.
--	---

VERSION 1 – AUTHOR RESPONSE

Daniel Allington and colleagues

Revisions requested and made:

1. Add 'cross-sectional' to title

Response: This has been done.

2. discuss more broadly the various mechanisms by which the observed disparities – particularly between racial / ethnic groups – may arise, beyond those examined here (i.e. via healthcare experiences and trust).

a. vaccine confidence is only one of the factors that affect vaccination uptake. For instance, Razai, Osama, and McKechnie describe concerns about access barriers, including locations of vaccine delivery, time of availability, residential segregation as a form of systemic racism (BMJ 2021;372:n513). Access barriers are also cited by the UK SAGE ethnicity sub-group (“Factors influencing COVID-19 vaccine uptake among minority ethnic groups, 17 December 2020”), Response: The Razai, Osama, and McKechnie article was already referenced with regard to its primary claim, i.e. that heightened levels of coronavirus vaccination hesitancy among minority ethnic communities may be attributable to reduced trust, in turn driven by experiences of racism, discrimination, and exclusion. (Providing an empirical test of this claim of Razai, Osama, and McKechnie’s was one of our primary motivations in carrying out the study, the other being an attempted replication of Freeman et al’s surprise finding that socio-demographic variables are not predictive of vaccine uptake.) Reference to Razai, Osama, and McKechnie’s mention of access barriers as an additional factor has now been added to the conclusion. It has additionally been possible to raise the issue of access barriers by reference to a systematic literature review which was published a few months after the initial submission of this article (Kamal et al 2021). This has an advantage over the Razai, Osama, and McKechnie study in that, being published later, it was able to refer to an empirical (although apparently not peer reviewed) study which found reported access barriers to have some degree of impact on vaccine uptake within minority ethnic communities. It is covered in the introductory section.

b. more broadly consider the range of potential drivers for such disparities when discussing these findings. For instance, the authors do mention in the introduction (i.e. page 5, lines 23-27) that racism, discrimination, and exclusion have the potential to contribute to vaccine hesitancy and circulation of misinformation, and through this mechanism be associated with decreased vaccine uptake. Beyond this one mechanism, however, I would strongly suggest that the authors engage more broadly – in both the introduction and discussion - with the variety of potential ways that historical and present racism may contribute to differential vaccination rates (for COVID and non-COVID vaccines) or access to similar health services.

Response: We have made an effort to emphasise that there are further potential mechanisms which may be at play, including access barriers. However, there is a limit to how broadly or deeply we can engage with alternative explanations, given the tight word limit specified by this journal, which the requested revisions have already pushed us far beyond. This is not a theoretical article attempting to

provide a full account of all the mechanisms which might potentially explain differential vaccination rates, but an empirical article testing one particular mechanism which has been put forward in published articles. Its greatest importance is in (a) the non-replication of Freeman et al's finding that socio-demographic variables have no predictive power with regard to vaccine uptake and (b) the support which it finds for Razai, Osama, and McKechnie's argument that ethnic disparities are at least partly attributable to a lack of trust resulting at least partly from worse experiences with healthcare. Of these two findings, the former provides evidence for the importance of ethnicity (in the face of a contrary finding published in a high impact journal), while the latter provides evidence of a need to search and test for further potential mechanisms beyond those considered in this study, potentially including access barriers. If the publication of this article can be permitted, it may perhaps inform future review articles in this way. In particular, we are concerned by the potential for Freeman et al's finding to lead to reduced emphasis on ethnicity in such review articles if correctives such as our own are not published.

c. on page 15 the authors write, "However, the fact that so much variance remains unexplained is in itself a cause for concern, and highlights the urgent need for further research to test the mediating role of other variables. Possible candidate mediators involve sources of information, conspiracy beliefs about the novel coronavirus, and attitudes to vaccination in general".

i. First, I might suggest that the authors highlight the urgent (broader) need to understand and resolve disparities in vaccination coverage, rather than focusing so narrowly on the need for research to test various potential mediating variables. While statistical analyses like these are useful, during a rapidly evolving vaccine roll-out programs may require the use of more qualitative information, for instance, to respond in real-time to emerging disparities.

Response: A sentence to 'highlight the urgent (broader) need to understand and resolve disparities in vaccine coverage has been added', along with two references to the need for qualitative studies, one in the introductory section and one in the discussion.

ii. Second, the authors highlight some potential reasons why their models can only explain some of the relationship between uptake and ethnicity. While "conspiracy beliefs" or attitudes towards vaccination in general may play a role, I think that it is important to contextualize these more fully. As above, might differential access play a role in these gaps? What factors might lead members of other-than-white ethnic groups to have different attitudes towards vaccination in general and/or hesitancy about a new vaccine (systemic racism and discrimination, legacies of previous unethical research, underrepresentation of racial/ethnic groups in medical research, whether as participants or researchers, etc.)?

Response: References to differential access has been added (see above). Reference to a study referring to unethical research practices etc (then in press, but now published) was already made in the introduction, and has now been slightly expanded in order to emphasise that this point was covered.

iii. Second, the authors note that comparison with official statistics suggests "slight representation of white ethnic groups" (pg 9, line ~ 38 – is this a typographical error for "overrepresentation"). It would be useful to have more detail to judge how representative this sample is (or is not) with respect with ethnicity within the UK. For instance, pg 7 / lines 6-10 seems to suggest that pre-stratification was used to produce representativeness of the UK adult population by age, gender, geographic region, and working status – but not by ethnicity. Is that true? If there was no pre-stratification or other effort in the sampling design to ensure that the sample was representative by ethnicity, are there any other data that would help the reader judge whether or not this population was likely to be representative by ethnicity? For instance, is there any information available on non-response rates by ethnicity? Were there any trends in the missingness mentioned in table 3.1 (by ethnicity or otherwise), or was this missingness random?

Response: Thank you for pointing out the typographical error. Please note that over-representation of white groups vis-a-vis the general population was only a feature of included cases, and not of the sample as a whole, and was a likely consequence of the substantially older age profile of white ethnic groups in the UK (in context of a policy of offering vaccination to UK residents in reverse order of

age). A table has now been provided showing representation of all ethnic groups within the sample that was collected. A note has also been added to the effect that pre-stratification by age, gender, geography, and working status but not ethnicity is standard in the British polling industry. Although we do not wish to engage in a lengthy discussion of sampling methodology, we note that there would be a strong argument that, in a context where almost all ethnic groups are very small, the use of ethnic quotas or pre-stratification would distort the sampling process to such an extent as to produce a less representative sample overall. There is also now a discussion of ethnic patterns in missingness.

iv. Related: the authors on page 5 lines 29-30 write that one of the strengths of the study was “representativeness of the UK adult population” - but that conflicts with the limitation in the third bullet point about limited representation of members of other-than-white ethnic groups. It seems that this statement about representativeness of the UK adult population should be revised in this light.

Response: It is stated that one of the strengths of the study is that it was done using a sample ‘designed for representativeness’. This is not contradicted by the statement that there was limited representation of other-than-white ethnic groups. In order to get a large percentage of other-than-white respondents, we would have had to create an unrepresentative sample through deliberate over-sampling. This point has now been clarified.

v. First, please include additional detail about how ethnic group was ascertained. Was this via self-report? If so, how was the question asked (as this may presumably influence the nature of the responses)? Was a post-hoc classification used (the footnote in Table 4.1 seems to indicate as much)? If so, how was this post-hoc grouping determined – by the authors, or was another standard classification applied? Were individuals allowed to choose more than one racial/ethnic category? If so, how were those individuals classified in this study?

Response: As the paper now explains, standard UK census ethnic categories were used (this was already implicit in the footer to table 4.1, but is now stated outright), and, as in the UK census, these were applied by self-report. (Given that the questionnaire was completed online, there is no other way in which ethnicity could have been determined than by self-report.) These standard categories were then dichotomised, as explained.

vi. Third, the analysis dichotomizes ethnic groups into “white” or “other than white”. I appreciate that this may have been a practical measure necessitated by sample size. However, this grouping homogenizes the diverse experiences of different other-than-white ethnic groups in the UK. At a minimum, this should be discussed as an important limitation of the study. In addition, it would be helpful if the authors could include information – if available – on which “other than white” ethnic groups were represented in their data set and if this distribution is reflective of the general UK population.

Response: The paper now discusses this limitation, and a detailed breakdown of ethnic categories in the sample is now both provided and discussed.

3. There are several study limitations that I think should be mentioned or discussed in more detail.

a. In addition, it seems like a significant limitation that – at the time of the survey – the vaccine had been available to older age groups for a longer period of time than it had for younger age groups. The association between age and uptake may therefore represent – at least in part - an artifact of the longer time of availability at the time of the survey. I would encourage the authors to add this as a potential limitation to the analysis, if thought to be applicable giving the timing of vaccination rollout by age and this data collection.

Response: This potential limitation has now been acknowledged in the limitations section; however, other studies which have found a negative association between age and vaccine hesitancy are now mentioned in the discussion section.

b. As the authors note, multiple logistic regression cannot fully eliminate the potential impact of this sort of a confounder. Did the authors consider any other sensitivity analyses to further investigate the interaction between age and ethnicity? It may be helpful to at least provide some additional description of the differences in age between ethnic groups. For instance, a negative correlation between age and membership in an other-than-white ethnic group. It would be useful, however, to report more clearly what the distribution of age was in respondents who identified as white vs those

who identified as other-than-white (ideally contextualizing this with respect to which age groups had access to the vaccine at which time). If the distributions are largely similar, at least with respect to COVID risk and vaccine availability by age, then the confounding may be less of an issue. If members of other-than-white ethnic groups were substantially younger, and the younger surveyed individuals in these groups tended to have lower vaccine uptake, then it is harder to interpret the findings of this study.

Response: This is true, but multiple regression remains the standard and arguably the best means by which to minimise the impact of confounds. The journal's explicit preference for brevity prevents a detailed discussion of these matters, but tables have been added containing age profiles of white and other-than-white groups. We note that this difficulty was already acknowledged in the final section of the article, and is now additionally acknowledged elsewhere in the article.

c. Last, my impression is that this is one of the major limitations of the study and should be included in the "strengths and limitations" bullet points on page 5.

Response: This has now been acknowledged with an additional bullet point.

4. Last, while the methods generally appear to be appropriate, there are opportunities to increase the clarity of the writing so that readers can better understand what exactly these data and analyses can – and can't – tell us.

a. First, I would suggest expanding the methods section – especially section 3.3, "Statistical Analysis". In particular, more detail about exactly what mediation analyses were performed and why would be useful. Some of these details are in the results section, but in general I was not sure which mediation analyses were performed, how they were selected, and what specific methods were used.

Response: We have now added some text clarifying the purpose of the mediation analyses as an exploratory study following on from the findings of the confirmatory analysis, and informing the reader that further information is available later in the paper. We must respectfully ask the reviewer to bear in mind that our ability to expand any section of the article is constrained by the 4000-word limit, which revisions have already pushed us a considerable distance beyond. All of the details requested are in the results section, and, given this journal's explicit preference for brevity, it would seem redundant to repeat all this information in the methods section.

b. Second, I appreciate that the authors clearly lay out their hypotheses. In reading the results, however, it is easy for the reader to get lost in the terminology (H1 vs H4, etc.) and the "rejection of the null hypothesis" phrasing. I found myself often having to refer back to the numbered list of hypotheses to understand what the writing meant. I would encourage the authors to more clearly state their findings in plain rather than statistical language and refer to hypothesis numbers sparingly if at all. It is also confusing to see that, in some cases, full models and in other cases partial models are used to evaluate the various hypotheses. For instance, on page 13, the authors state (using a full model) "Thus, the null hypothesis can be rejected with regard to H6, but not with regard to H7." But the partial model including just NHS experiences and demographic predictors shows a significant effect for NHS experiences. And, earlier in the section (lines 45-46, page 12) the authors test a number of hypotheses using the demographics-only model. Essentially, there are many models being run, and it is unclear to me how the authors are choosing which model to use to test which hypotheses in some cases.

Response: It is true that there were many models, which creates the potential for confusion. However, the decision to report all models in full was made in the interests of full transparency (and in accordance with the pre-registration document). The methodology employed was to test all of the hypotheses using all of the models in which the relevant variables were employed, but to consider the full model to be the definitive test. This point has now been clarified.

The point that the partial model including just NHS experiences and demographic predictors shows a significant effect for NHS experiences was itself the motivation for the exploratory mediation analysis; this has also been clarified.

We note that frequent reference back to the numbered list of hypotheses is expected when reading a complex quantitative study of this kind, and that spelling the hypotheses out in full rather than using numbers would increase the word count still further. As explained above, the purpose of this article is not to provide an introduction to or overview of the subject area but simply to report – as succinctly as possible – the findings of a single quantitative study which may then be summarised in review articles such as Kamal et al (2021). With respect, we ask the reviewers to bear this in mind when deciding whether or not to permit publication of this article.

5. Minor comments:

a. I am not sure what the sentence on page 5 lines 37-47 means: “Moreover, it would appear plausible that the range of mechanisms which can be assumed to generate the forms of exclusion associated with racism might also be associated with decreased trust, more negative experiences of healthcare, and higher rates of vaccine hesitancy among other groups, including women, young people, less highly educated people, and members of low income households”. In particular, I am not sure what the “range of mechanisms which can be assumed to generate the forms of exclusion associated with racism” is meant to represent in this context. Is there a way to phrase this more clearly?

Response: We agree that this sentence was rather confusing; it has now been re-written.

b. Page 7 lines 42-44: I'm a bit confused by how gender was ascertained. It seems here that survey participants were asked to categorized themselves as “female” or “other than female”. However, the text goes on to say that all individuals identified as either male or female. How was this information ascertained? Perhaps I'm missing something – did the actual survey include more gender groupings (or free response, etc) and then the authors re-categorized into female and other than female?

Response: The actual survey included an additional gender grouping, but it was not used by any respondents; however, the recategorisation was included in the pre-registration document and therefore had to be carried out even though it was unnecessary. This is now clarified.

c. Table 4.1: Are some of these values in the table percentages? If so, would clarify either in the cells themselves or in the headers. Also, please ensure that all abbreviations used in table headers are defined.

Response: Percentage symbols have been added to the cells themselves. Abbreviations are now defined in the table footers.

d. Table 4.1 and 4.2; page 9 / line 33: Please explain in the methods how the weights used for these weighted analyses were derived.

Response: It is RIM weighting that was used. This is now stated in the paper.

e. Page 9 line 39: should “representation” be “overrepresentation” here?

Response: Yes. This error has now been corrected.

6. Reviewer 2 comments:

a. in the title (page 3), would you please kindly include the full form of NHS?

Response: This has now been done.

b. in the section study design (page 6), in line 35, would you please kindly explain how did you choose top two levels recorded as true and other recorded as false?

Response: As the paper now explains, this was an arbitrary choice. (The split had to be made somewhere, and it would not have been possible to recode the top three levels consistently as not all ordinal variables used had more than three levels. An alternative would have been to recode only the top level as true and the others as false; in the one instance where we believe that this could have made a difference, i.e. household income, this was already discussed in the paper.)

c. in the section statistical analysis (page 7), in line 50, would you please kindly explain why are you taking 10,000 repetitions?

Response: There is no standard number of repetitions for bootstrap analysis; however, the number used here is far in excess of the software default 1000 repetitions. The greater the number of repetitions, the less the resulting confidence intervals are subject to random variation, but the more time it takes for a given computer to complete the calculations. This point is now made in the paper.

d. in the result section (page 8), in table 4.1, would you please kindly explain all the numbers are in percentage?

Response: Percentage symbols have been added to the table cells.

e. in the power analysis section (page 8), in line 1, would you please kindly explain how you find the sample size of 2773 for power calculation?

Response: The words 'i.e. the number of observations which were sufficiently complete to be fitted in the full model' have been added in order to make explicit the point that this was the actual N.

f. in the result section (page 9), in table 4.3, would you please kindly explain how do you select the variables for bivariate correlations?

Response: Bivariate correlations between all variables in the study were calculated and reported, so there was no selection. The word 'all' has been added to the sentence beginning with the words 'Table 4.3 shows' in order to make this point explicit.

g. in the discussion section (page 14), you discussed only the result of your study but there is no comparison with the published study's result in UK or other countries.

Response: All other relevant published studies available at the time of writing had already been discussed in the introductory section. At this point in time, no other published studies have sought to measure the mediating effect of healthcare experiences on the relationship between ethnicity and vaccine uptake. We have, however, been able to include reference in the introductory section to a review article which had not been published at the time of this article's original submission, and which covered many other studies (Kamal et al 2021). We hope that the reviewers will consider this sufficient.

VERSION 2 – REVIEW

REVIEWER	Mosser, Jonathan University of Washington, Health Metrics Sciences
REVIEW RETURNED	19-Jan-2022

GENERAL COMMENTS	I thank the authors for their thoughtful revisions to the original manuscript. I acknowledge that I provided a large number of queries and comments, and appreciate the effort required to respond to all of these. The revised manuscript is, in my opinion, greatly strengthened by these revisions, and I hope that the authors share this view. All of my comments have been adequately addressed and I have no further comments aside from a few typographical errors that I happened to noticed (which are listed at the end of the review). At this point, in my view, the article would be well suited for publication, should the editorial team agree. Please find my responses to the authors' revisions below, organized by comment number from the original submission. Major comments: 1. Regarding a broader discussion of the potential drivers of vaccination uptake and racial/ethnic disparities observed in the manuscript. I appreciate the addition of the Kamal et al article and inclusion of access barriers as an additional potential driver, and more generally the discussion of the potential mechanisms for the observed discrepancies strengthens the paper. The additional emphasis on contextualizing drivers of mistrust (e.g. awareness of past mistreatment and unethical practices and current perceptions of racism) is also appreciated and provides valuable context to frame this discussion.
---

	I also appreciate the authors' response that the paper's scope focuses on a narrow subset of mechanisms that may drive these discrepancies, and that a full review of such mechanisms is outside of the scope of this paper. However, at least mentioning the existence of further potential mechanisms (such as access barriers) helps to ensure that the reader is aware of other plausible pathways that may contribute to such discrepancies. The authors have now nicely struck this balance. Similarly, I appreciate that the authors have broadened the piece's perspective on the types of data and approaches that may be useful to resolve these discrepancies, including by acknowledging the complementary utility of qualitative research and adding language that highlights the need for action in addition to academic study. I have no further comments in this regard and greatly appreciate the authors' efforts to broaden these discussions. 2. Related comments regarding classification of ethnic groups and representativeness: a. How was ethnic group ascertained? I appreciate the additional clarity provided by the authors regarding self-report and the use of categories from the UK census categorization. No additional comments. b. Regarding the slight overrepresentation of white ethnic groups. I thank the authors for the additional clarity provided by their revisions. Table 3.1 is particularly helpful as a means to better understand the composition of the white and other than white dichotomized groupings, and I appreciate the addition of Table 3.2 along with the discussion of missingness. In all, the revised manuscript allows the reader to have a much more thorough understanding of the data used in the analysis. No further comments. c. Regarding dichotomization of ethnic groups for analysis. I appreciate both the additional clarity provided by Table 3.1 and the addition of a sentence that acknowledges the limitations inherent in dichotomizing ethnicity – many thanks. No further comments. 3. Regarding additional limitations to consider a. Regarding the association between time since availability of vaccine and age. The authors have adequately addressed this by adding this limitation to the "Limitations" section. No further comments. b. Regarding age as a confounder of the relationship between ethnicity and vaccine uptake. The author's comments are well received, and on re-reading the relevant text of the limitations section (as well as the addition of age profiles of white and other-than-white ethnic groups and highlighting this limitation in the bullet points at the start of the article), I feel that this point has been well addressed. No further comments.
--	---

	4. Regarding the methods section a. Further explanation of mediation analysis methodology. The brief additions to the “Statistical analysis” section help greatly to clarify the model and set the stage for the subsequent results. I thank the authors for the addition; no further comments. b. Style of reference to multiple hypotheses by number. I appreciate the authors’ comments regarding pre-registration of these analyses and the selection of the primary (full) model. In addition, the clarified discussion on lines 30-39 of page 42 in the combined PDF helps provide a clear interpretation of the key findings. I do still find the results somewhat hard to follow, requiring multiple references back to the table of listed hypotheses. This may simply be a stylistic preference, however, so I defer to the editor in this regard. Minor comments:  1. Regarding clarification of a lengthy sentence (original page 5, lines 37-47). The revised language is much clearer – thanks to the authors for the revision and no further comments. 2. Regarding the classification of “female” vs “other than female”. Similarly, the revised language is much clearer; no further comments. 3. Regarding Table 4.1: The table is much more easily interpretable with percentage symbols and clear definition of the abbreviations; no further comments. 4. Regarding the generation of weights: The added text in the second paragraph of the Data Collection section is much appreciated; no further comments. 5. Regarding a typographical error (representation vs overrepresentation). Thanks to the authors for correcting this error, no further comments. Additional minor comments: I encountered a few typographical errors in the revised manuscript, which I mention here for the authors’ attention:  • Line 25 on page 5 (Introduction) “hesitancyuptake” should probably be one or the other • Line 54 on page 15 (Confirmatory analysis): “Mroreover” should be “Moreover”
--	---

REVIEWER	Tabassum, Mimma Noakhali Science and Technology University, Statistics
REVIEW RETURNED	02-Feb-2022

GENERAL COMMENTS	I recommend for accept the manuscript.
--